

# Delocalization transition in low energy excitation modes of vector spin glasses

Silvio Franz[1], Flavio Nicoletti[1,2], Giorgio Parisi[2,3] and Federico Ricci-Tersenghi[2,3]

**1** LPTMS, UMR 8626, CNRS, Univ. Paris-Sud, Université Paris-Saclay, 91405 Orsay, France
**2** Dipartimento di Fisica, Università "La Sapienza", P.le A. Moro 5, 00185, Rome, Italy
**3** INFN, Sezione di Roma1, and CNR–Nanotec, Rome unit, P.le A. Moro 5, 00185, Rome, Italy

## Abstract

We study the energy minima of the fully-connected $m$-components vector spin glass model at zero temperature in an external magnetic field for $m \geq 3$. The model has a zero temperature transition from a paramagnetic phase at high field to a spin glass phase at low field. We study the eigenvalues and eigenvectors of the Hessian in the minima of the Hamiltonian. The spectrum is gapless both in the paramagnetic and in the spin glass phase, with a pseudo-gap behaving as $\lambda^{m-1}$ in the paramagnetic phase and as $\sqrt{\lambda}$ at criticality and in the spin glass phase. Despite the long-range nature of the model, the eigenstates close to the edge of the spectrum display quasi-localization properties. We show that the paramagnetic to spin glass transition corresponds to delocalization of the edge eigenvectors. We solve the model by the cavity method in the thermodynamic limit. We also perform numerical minimization of the Hamiltonian for $N \leq 2048$ and compute the spectral properties, that show very strong corrections to the asymptotic scaling approaching the critical point.

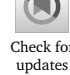

## Contents

Low energy excitations of glasses display a remarkable degree of universality. In addition to usual phonons and other extended modes, in a variety of model glassy system it has been found the presence of low energy quasi-localized excitations with density of states (DOS) behaving quartically at low frequencies $D_{\text{QLS}}(\omega) \sim A\omega^4$ [1–11]. While the prefactor is found to depend on the details of the models and preparation protocols [10, 11], the $\omega^4$ behavior seems to be very general, independent of composition, preparation procedure and the space dimension [8, 9, 12]. Remarkably, the same $\omega^4$ behavior of the DOS can be also found in a granular amorphous solid with long-range electroscatic interactions [13]. Despite the fact that this spectrum of localized modes was first predicted by phenomenological theories [14, 15], a theoretical comprehension based on microscopic models, as well as an understanding of its generality is at present lacking.

In the theoretical study of glassy landscapes it is useful to turn to mean-field models, where the Hessian of typical minima are random matrices from the classical Wigner-Dyson ensemble, or simple variations. Mean-field models usually display either a gapped spectrum or a quadratic DOS, $D(\omega) \propto \omega^2$, at low frequencies [16, 17]. In both cases, the corresponding eigenstates are delocalized, since they are related to the eigenvectors of simple random matrices.[1] Only very recently a mean-field spin glass model has been studied [18–20] displaying, in presence of an external field, $D(\omega) \propto \omega^4$ with localized modes. However, this model has unbounded variables subject to a constraining quartic potential, very useful for performing computations, but rather unrealistic. We aim at studying spin glasses with continuous degrees of freedom, also known as vector spin glasses, where every variable is a fixed norm vector of $m \geq 2$ components. The most common models consider XY variables ($m = 2$) and Heisenberg variables ($m = 3$).

Vector spin glasses are good candidates for glassy models having a non-trivial spectrum of low-energy excitations. In Ref. [21] the Hessian of the minima of the Heisenberg spin glass model in three dimension in presence of a magnetic field was considered. It was found numerically that the model has a zero temperature phase transition as a function of the field intensity, from a paramagnetic phase with a single isolated disordered minimum to a spin-glass phase with many minima. The paramagnetic phase has remarkable features: one finds that within this phase, it exists a value of the field, below which the spectrum of the Hessian at the dominating energy minimum become gapless, and its properties approach the one found in glasses. The low frequency behavior of the density of states behaves as $\omega^4$ and corresponds to quasi-localized modes of excitation. The behavior changes at the spin-glass transition, and strong hints were found that the low lying eigenvectors become delocalized.

Recent studies showed a similar picture for the XY spin glass model on the Bethe lattice in a field, which also has a paramagnet to spin-glass transition at zero temperature varying the external field. The model can be solved analytically with the cavity method [22] and thus the phase diagram can fully determined [23, 24]. Also in this case, one finds the absence of a gap in the spectral density, a density of low-energy excitations behaving as $D(\omega) \propto \omega^4$ and the quasi-localized nature of the low energy excitations [25].

---

[1]The relation between eigenvalues and frequency $\lambda = \omega^2$ implies $\sqrt{\lambda}\,d\lambda \sim \omega^2 d\omega$.

Unfortunately the solution of models on the Bethe lattice is rather involved and one would like to find similar results in mean-field models defined on fully connected graphs. Among the latter the most studied have been spherical models: $p$-spin models, that are paradigmatic for the random first order transition, and perceptrons, allowing to formulate the simplest model for jamming. In spherical $p$-spin models the structure of the minima of the energy has been studied extensively [26]. One finds either a paramagnetic phase with a single minimum, or a stable glass phase with many, well separated, stable minima, or a marginal glass phase with marginal minima that lie one close to each other in the energy landscape. The Hessian matrices in the different minima, up to an overall shift of the diagonal elements, turn out to be random matrices from the Wigner-Dyson (GOE) ensemble. Correspondingly, the lower spectral edge presents a square root singularity, typical of dense random matrices. The spectral gap, determined by the overall shift, is positive both in paramagnetic and stable glass minima while it vanishes in marginal glass minima. In the perceptron, where the Hessian is a shifted Wishart random matrix the situation is similar, but richer due to the presence of a jamming point, where the spectral density behaves as $\lambda^{-1/2}$ in the origin [16].

The presence of a spectral gap in the stable glass and paramagnetic phases, seems to be a limitation of fully-connected models. Moreover, the dense nature of the Hessian matrix implies complete delocalization of the eigenvectors, that for the above-mentioned ensembles are just random points on the hypersphere defined by the spherical constraint, irrespective of the eigenvalue they correspond to.

In this paper we would like to reexamine these points for the richer models of vector spin glasses defined on fully-connected graphs. The study of these models were pioneered by Bray and Moore in ref. [27–29]. It is well known, that differently from the Ising and XY spin glasses, the fully-connected $m$-component vector spin-glass models with $m \geq 3$ have a zero temperature transition in a field [30]. In particular the Heisenberg fully-connected spin glass in a field at zero temperature has been studied in Ref. [17], where the authors find a gapped spectrum in the paramagnetic phase (i.e. for a large enough external field). We are going to revisit this result, showing that the spectrum of this model extends down to $\lambda = 0$ in the whole paramagnetic phase, with a pseudo-gap behavior $\rho(\lambda) \propto \lambda^{m-1}$.

The Hessian matrix of vector spin glass models is a random matrix of the Rosenzweig-Porter ensemble [31] (also called deformed Wigner-Dyson ensemble) with random diagonal elements whose statistics is directly related to the distribution of local fields. We present a derivation of the spectral properties of the Hessian matrix based on the cavity method, that returns the known result [32] with a very clear physical interpretation.

We then study the eigenvectors, whose inverse participation ratio turn out to be proportional to $1/N$ with a prefactor that diverges on the edges of the spectrum as $\lambda^{-2(m-1)}$, revealing non-trivial localization properties even in the paramagnetic phase of fully-connected mean-field spin glass models, induced by the random external field.

Approaching the critical field, where a paramagnet to spin glass transition takes place, the (quasi-)localized low-energy eigenmodes undergo a delocalization transition, becoming system-wide extended. Below the critical line, in the spin glass phase, we expect the spectrum of the Hessian to have a square root singularity and extended eigenvectors.

To support and complement the analytical results obtained in the thermodynamical limit, we perform energy minimization for a large number of samples of sizes up to $N = 2048$, and we compute the spectrum of the Hessian at the energy minimum. The numerical data fully support the scenario obtained analytically, but also reveal the presence of large finite size corrections, that become even larger approaching the critical point, possibly hiding the correct asymptotic scaling.

# 1 Analytical solution of the model

## 1.1 The cavity equations

In this section we review some well known properties of the fully-connected $m$-component vector spin glass at zero temperature in an external random field. The model is defined by the Hamiltonian

$$\mathcal{H} = -\sum_{i<j} J_{ij} \mathbf{S}_i \cdot \mathbf{S}_j - \sum_i \mathbf{b}_i \cdot \mathbf{S}_i \,, \tag{1}$$

where the $N$ spins $\mathbf{S}_i$ are $m$-components vectors, normalized to $|\mathbf{S}_i| = 1$, and the couplings $J_{ij}$ (with $i < j$) are Gaussian independent and identically distributed random variables (iidrv) with $\overline{J_{ij}} = 0$ and $\overline{J_{ij}^2} = 1/N$. For $i > j$, $J_{ij} = J_{ji}$, while $J_{ii} = 0$. We choose the external fields to be iidrv with Gaussian distribution of zero mean and variance $\overline{(b_i^\alpha)^2} = \Delta^2$. At zero temperature, the Gibbs measure concentrates on the absolute minima of $\mathcal{H}$, where the following equations hold

$$-\mathcal{H}_i' \equiv -\partial_{\mathbf{S}_i} \mathcal{H} = \sum_j J_{ij} \mathbf{S}_j + \mathbf{b}_i = \mu_i \mathbf{S}_i \,, \tag{2}$$

where $\mu_i = |\mathcal{H}_i'|$. Eq.(2) expresses the fact that in minima the spins are oriented along their local fields.

This set of equations can be analysed with the cavity method. As usual one compares the solution of the full system of equations with $N$ spins, with the the solution of a system where a single spin $i$ is removed. The crucial hypothesis is that the solutions of (2) are continuous when the spin is removed. Since the couplings are small, the effect of a single spin $i$ on the others is small and can be treated within linear response.

If the system is in a replica symmetric phase, there is a single relevant low-energy solution that can be followed straightforwardly. If replica symmetry is broken there are multiple solutions of (2) that are quasi-degenerate with the ground-state: the removal or addition of one spin causes level crossing which should be taken into account when one is interested to the statistics of the solutions.

Denoting by $\mathbf{S}^{(i)}$ the solution of the minimization problem in absence of $i$, and $\mathbf{S}$ the corresponding solution of the full problem, we can write [33]

$$\mathbf{S}_j = \mathbf{S}_j^{(i)} + \chi_{jj} J_{ji} \mathbf{S}_i \quad \text{with} \quad \chi_{jj} = \frac{\delta \mathbf{S}_j}{\delta \mathbf{b}_j} \,. \tag{3}$$

Notice that the susceptibility $\chi_{jj}$ is an $m \times m$ matrix. Inserting into Eq. (2) we find

$$-\mathcal{H}_i' = \sum_j J_{ij} \mathbf{S}_j^{(i)} + G_0 \mathbf{S}_i + \mathbf{b}_i \,, \tag{4}$$

where $G_0$ is given by $G_0 \mathbb{I}_m = \sum_j J_{ij}^2 \chi_{jj} = \sum_j \chi_{jj}/N$ and is thus the average diagonal component of the susceptibility matrix, $G_0 = \sum_j \chi_{jj}^{\alpha\alpha}/N$, which is obviously independent of $\alpha$. The variables $J_{ij}$ are independent from $\mathbf{S}_j^{(i)}$, so that the cavity fields $\mathbf{h}_i = \sum_j J_{ij} \mathbf{S}_j^{(i)}$ are Gaussian random variables with zero mean and covariance matrix $\overline{h_i^\alpha h_j^\beta} = \frac{1}{m} \delta_{ij} \delta_{\alpha\beta}$.

If the variance of the external field $\Delta$ is large enough, the system is replica symmetric and a single solution needs to be considered. The components of total cavity field $\mathbf{H}_i = \mathbf{h}_i + \mathbf{b}_i$ are

also Gaussian with variance $\sigma^2 = \Delta^2 + 1/m$, while the parameters $\mu_i$ are given by $\mu_i = G_0 + H_i$ with $H_i = |\boldsymbol{H}_i|$. As a consequence the moduli $H_i$ have distribution

$$P_m(H) = \frac{1}{Z_m} H^{m-1} e^{-H^2/(2\sigma^2)} , \tag{5}$$

where $Z_m = \Gamma(m/2)(2/m + 2\Delta^2)^{m/2}/2$. Notice that the variables $\mu_i = |\mathcal{H}'_i|$ verify the well known stability condition $\mu_i \geq G_0$ [33].

## 1.2 The Hessian

The excitations around the minima, are characterized by the spectrum of the Hessian matrix whose elements we write as $M_{ij}^{\alpha\beta}$. The Hessian matrix, around a minimum, restricted to spin fluctuations that keep each spin on its $m$-dimensional spheres of unit norm, can be written as

$$M_{ij}^{\alpha\beta} = \sum_\gamma P_i^{\alpha\gamma} P_j^{\gamma\beta} \left( -J_{ij} + \mu_i \delta_{ij} \right) , \tag{6}$$

$$P_i^{\alpha\beta} = \delta_{\alpha\beta} - S_i^\alpha S_i^\beta , \tag{7}$$

which is a symmetric random matrix. To understand its statistics, we observe that the dependence of the diagonal elements $\mu_i$ on each of the $J_{ij}$ is very weak, and it can be safely neglected. In the $N(m-1) \times N(m-1)$ space orthogonal to $\boldsymbol{S}$ we have then a Rosenweig-Porter random matrix, whose off-diagonal elements have a Gaussian distribution of zero mean and variance of order $1/N$, while the diagonal elements are finite random variables distributed like the $\mu_i$ variables. A more precise definition could be given introducing on each site $i$ an orthonormal basis of $m$-vectors $\{\boldsymbol{S}_i, \boldsymbol{u}_{i,1}, \dots, \boldsymbol{u}_{i,m-1}\}$. The Hessian is thus the $N(m-1) \times N(m-1)$ matrix given by $M_{ij}^{ab} = (-J_{ij} + \mu_i \delta_{ij}) \boldsymbol{u}_{i,a} \cdot \boldsymbol{u}_{j,b}$.

The properties of the Hessian can be studied with the cavity method, following a procedure similar to the one we have used for the equations defining the minima. We write the eigenvalue equations in presence of a small external source $h_i^\alpha$

$$-\sum_j J_{ij} v_j^\alpha + \mu_i v_i^\alpha - \lambda v_i^\alpha = h_i^\alpha , \tag{8}$$

where the index $\alpha$ runs over the $m$ components, the eigenvector with $mN$ components can be written as $\boldsymbol{v}(\lambda) = (\boldsymbol{v}_1, \dots, \boldsymbol{v}_N)$ and each $\boldsymbol{v}_i$ is orthogonal to the corresponding $\boldsymbol{S}_i$, i.e. $\sum_{\alpha=1}^m v_i^\alpha S_i^\alpha = 0$. A small imaginary part in $\lambda$ is implicitly assumed to insure invertibility.

As in the previous section, we single out a site $i$ and compare the solution of the full system (8) to the one where the site $i$ is removed. Defining $\boldsymbol{v}_j^{(i)}$ the solution of Eq. (8) in absence of spin $i$ and assuming continuity, we can write

$$-\sum_j J_{ij} \boldsymbol{v}_j^{(i)} - G(\lambda) \boldsymbol{v}_i + \mu_i \boldsymbol{v}_i - \lambda \boldsymbol{v}_i = \boldsymbol{h}_i , \tag{9}$$

where $G(\lambda) = \sum_j G_{jj}^{\alpha\alpha}(\lambda)/N$ is the mean value of the diagonal component of the $\lambda$-dependent susceptibility $G_{jj}^{\alpha\alpha}((\lambda) = \partial v_j^\alpha / \partial h_j^\alpha$ (all the other components of the susceptibility matrix are zero on average). This susceptibility is directly related via $G(\lambda) = \mathrm{Tr} R(\lambda)/(Nm)$ to the resolvent matrix, defined by

$$R_{ij}^{\alpha\beta}(\lambda) = \sum_{\gamma\delta} P_i^{\alpha\gamma} \left[ (\mathbb{J} + \mathrm{diag}(\{\mu_i\}) - \lambda \mathbb{I}_N)^{-1} \right]_{ij}^{\gamma\delta} P_j^{\delta\beta} ,$$

from which we get the spectral density by the usual limit

$$\rho(\lambda) = \lim_{\eta \to 0} \frac{m}{\pi(m-1)} \mathrm{Im}(G(\lambda + i\eta)) . \tag{10}$$

The prefactor $m/(m-1)$ takes into account that fluctuations are restricted to the directions orthogonal to the spins.

Taking the derivative of Eq. (9) w.r.t. $h_i^\alpha$ we get an equation for the local resolvent

$$G_{ii}(\lambda) = (1 - 1/m)(H_i + G_0 - \lambda - G(\lambda))^{-1} \,, \tag{11}$$

and averaging over $i$ the self-consistent equation for $G(\lambda)$

$$G(\lambda) = \frac{m-1}{m} \int dH \frac{P_m(H)}{H + G_0 - \lambda - G(\lambda)} \,, \tag{12}$$

$$\text{with} \qquad G_0 \equiv G(\lambda = 0) = \frac{m-1}{m} \int dH \frac{P_m(H)}{H} \,.$$

Knowing the cavity field distribution $P_m(H)$, Eq. (12) can be solved numerically and analysed analytically for small $\lambda$. giving us access the spectral density, once we separate the real, $G'(\lambda)$, and imaginary, $G''(\lambda)$, parts of $G(\lambda)$.

Notice that Eq. (9) gives us also access to the statistics of eigenvectors. The statistical properties of eigenvectors of the Rosenzweig-Porter ensemble have been recently discussed in [34] with supersymmetry, in [35] with Dyson Brownian motion and rigorously proven in [36]. The analysis via Eq. (9) offers a quick way of obtaining many of the results of these papers. One can easily realize that for $h_i^\alpha \to 0$ a non-vanishing solution to Eq. (9) is such that $\langle |v_i^\alpha|^2 \rangle \propto |\mu_i - \lambda - G(\lambda)|^{-2}$, where the angular brackets represent the average over all spins with a field $H_i = \mu_i - G_0$ and the normalizing constant should be fixed imposing $\sum_{i,\alpha} \langle |v_i^\alpha|^2 \rangle = 1$. Noticing that the imaginary part of Eq. (12) implies $(m-1)/m \int dH P_m(H)|H + G_0 - \lambda - G(\lambda)|^{-2} = 1$, we have

$$\langle |v_i^\alpha|^2 \rangle = \frac{m-1}{Nm^2|H_i + G_0 - \lambda - G(\lambda)|^2} \,. \tag{13}$$

With respect to the simple Rosenzweig-Porter ensemble we should take care to the fact that the $v_i$ should be orthogonal to the $S_i$. If not for that reason, we would immediately conclude that the components $v_i(\lambda)$ of the eigenvector corresponding to eigenvalue $\lambda$ are independent Gaussian random variables[2], with variances given by Eq. (13). We will discuss more this point at the time of computing the inverse participation ratio.

Notice that Eq. (13), while it implies $O(1/\sqrt{N})$ elements for the bulk eigenvectors, on the edge of the spectrum it admits solutions localized on a single site $i$ in correspondence of eigenvalues $\lambda$ of the Hessian such that $|H_i + G_0 - \lambda - G(\lambda)|^2 = O(1/N)$ for some $i$. In this case the component $i$ of the eigenvector $v(\lambda)$ is such that $|v_i(\lambda)| = O(1)$. We will find such solutions in the paramagnetic phase, with an amplitude that vanishes at the critical point.

## 1.3 The spin glass transition

The solution to Eq. (12) can be found in the paramagnetic phase, where the cavity field distribution $P_m(H)$ is given by Eq. (5). The paramagnetic solution is stable as long as the spin glass susceptibility is finite. This quantity is defined as

$$\chi_{\text{SG}} = \frac{1}{Nm} \sum_{ij\alpha\beta} \left( \frac{\partial S_i^\alpha}{\partial b_j^\beta} \right)^2 = \frac{dG}{d\lambda}\bigg|_{\lambda=0} = \frac{A}{1-A} \,, \tag{14}$$

$$A = \frac{m-1}{m} \int dH \frac{P_m(H)}{H^2} = \frac{(m-1)}{(m-2)(1+m\Delta^2)} \,.$$

---

[2]We neglect the small dependence between components due to overall normalization of the eigenvectors.

We find therefore the well known condition $A < 1$ [27], which is verified for $\Delta > \Delta_c = \frac{1}{\sqrt{m(m-2)}}$. At $\Delta_c$ we have $A = 1$ and $\chi_{SG}$ diverges as $\chi_{SG}(\Delta) \propto (\Delta - \Delta_c)^{-1}$, as expected in mean field models. Hereafter we consider only $m \geq 3$ values, as for $m = 2$ the system is in the spin-glass phase for all values of the field.

## 1.4 The lower edge of the Hessian spectrum

From Eq. (14) it is simple to see that Eq. (12) for the resolvent is incompatible with $G''(\lambda) = 0$ for small positive $\lambda$. We postpone to the Appendix A and B a detailed analysis and the derivation of the several different solutions depending on whether the system is in the paramagnetic phase ($\Delta > \Delta_c$) or at the critical point ($\Delta = \Delta_c$), and also depending on the value of $m$. Here we report a summary of these results, which turn out to be well compatible (apart some logarithmic corrections at specific values of $m$) with those derived in Ref. [32] for deformed Wigner matrices.

We introduce a parameter $\epsilon = 1 - A$ setting the distance from the critical point. It can be shown that this parameter coincides with the replicon eigenvalue setting the stability of the replica symmetric solution in the replica formalism [27]. While in the following $\epsilon$ will not have to be necessarily considered small, close to the critical point it takes the form

$$\epsilon = \frac{2\sqrt{m(m-2)}}{m-1}(\Delta - \Delta_c) + o(\Delta - \Delta_c). \tag{15}$$

For $\epsilon > 0$, in the paramagnetic phase dominated by the external fields, the density of states at the lower band edge of the Hessian spectrum is controlled by the distribution of the cavity field that behaves as $P_m(H) \propto H^{m-1}$ for small values of $H$. As a consequence we have

$$\rho(\lambda) = \frac{1}{\pi}\frac{m}{m-1}G''(\lambda) \simeq \frac{1}{\epsilon}P_m(\lambda/\epsilon) \propto \frac{\lambda^{m-1}}{\epsilon^m}, \tag{16}$$

while for the real part of $G(\lambda)$ we get

$$G'(\lambda) \simeq \chi_{SG}\lambda = \frac{1-\epsilon}{\epsilon}\lambda. \tag{17}$$

As announced the spectrum is ungapped for all values of the field. Notice that the condition $(1 - 1/m)\int dH \frac{P(h)}{|H+G_0-\lambda-G(\lambda)|^2} = 1$, which should be valid for all $\lambda$ in the support of $\rho(\lambda)$ and expresses the normalization of the eigenvectors (13), cannot be fulfilled for $\lambda \to 0$ if $A < 1$. Eigenvectors corresponding to the smallest eigenvalues should therefore condense a finite fraction of weight on a single component, with a mechanism similar to the Einstein condensation of the Bose gas (see below for more details).

Approaching the critical point the $\epsilon$ dependent coefficient in Eq. (16) diverges and indeed the low-energy excitations become more and more abundant. Exactly at criticality ($\epsilon = 0$) the shape of the lower band edge does depend on the specific value for $m$ as follows

$$\rho(\lambda) \propto \sqrt{\frac{\lambda}{|\log\lambda|}} \qquad \text{for } m = 3, \tag{18}$$

$$\rho(\lambda) \propto \sqrt{\frac{\lambda}{J}} \qquad \text{for } m \geq 4, \tag{19}$$

where $J \propto \int dH P_m(H)H^{-3}$. The spectrum for small $\epsilon$ has a crosses-over from the $\rho(\lambda) \sim \lambda^{m-1}/\epsilon^m$ behavior at $\lambda \ll \lambda^*$ to a square root behavior $\rho(\lambda) \sim \sqrt{(\lambda - \lambda^*)/J}$ for $\lambda \gtrsim \lambda^*$. The cross-over eigenvalue $\lambda^*$ is estimated in Appendix C to be $\lambda^* \sim \epsilon^2/J$. For $m = 3$ the mean of $H^{-3}$ is divergent and a logarithmic corrections should be expected.

Let us comment on the above findings. First of all, at variance with previous studies [17], we find that the spectrum of the Hessian of fully-connected vector spin glasses is gapless in the paramagnetic (high field) phase. The spectrum shows a pseudo-gap, $\rho(\lambda) \propto \lambda^{m-1}$, induced by the probability law of cavity fields that determine the diagonal elements of the Hessian. The density of low cavity fields $P_m(H) \propto H^{m-1}$ is just a consequence of the statistical rotation invariance together with the fact that the cavity fields are just Gaussian vectors in the paramagnetic phase. In other words, the presence of low energy excitations is a simple consequence of the disordered nature of the minima and of the abundance of small fields. The factor $H^{m-1}$ is just a local entropic term determined by the nature of the microscopic variables. One can easily imagine very different forms for such local terms depending on the system under study. We also notice that the pseudo-gap disappears for $m \to \infty$ corresponding to the spherical model. We remind that the result $\rho(\lambda) \propto \lambda^{m-1}$ translates to a low-frequency DOS, $D(\omega) \propto \omega^{2m-1}$. This is different from what happens in disordered minima of finite dimensional glassy models, where a $\omega^4$ behavior seems to be ubiquitous.

A DOS of quasi-localised modes that depends on the dimension of the site variable can be found also in [37]. In that work the authors studied the lower band edge spectrum of a mean field model of soft spheres, such that the contact network of the spheres is a Bethe lattice, but the average number of contacts $\bar{z}$ depends on the dimension $d$ of the contact vectors. In this aspect, it is very similar to the present vector spin glass model with $d$ components, which is a mean-field model with finite-dimensional spin variables. In the hyperstatic phase $\bar{z} > 2d + 1$ they found $D(\omega) \sim \omega^{\alpha(d)}$, i.e. an exponent depending on $d$.

On approaching the spin glass transition the spectrum has a crosses-over from the $\lambda^{m-1}$ behavior to a square root behavior, which in terms of frequencies means a DOS behaving as $\omega^2$. The cross-over occurs around a characteristic value $\lambda^* \sim \epsilon^2$. We notice here a similarity of behavior with simulations of structural glasses, where the $\omega^4$ behavior of quasi-localized excitations crosses-over to a $\omega^2$ behavior of extended excitations.

The above cavity computation cannot be extended to the spin glass phase straightforwardly, since the detailed knowledge of the cavity field distribution would need to take into account Replica Symmetry Breaking effects, the presence of many dominating energy minima and the marginality of each of them. Nonetheless, we know that the spin glass susceptibility is divergent in the whole $\Delta < \Delta_c$ space phase, and this is enough to conclude that spectral density should display a simple square root behavior in the whole spin glass phase for all $m$ (see Appendix for a detailed analysis).

## 1.5 Localization at the lower edge

In dense matrices, the bulk eigenvectors are generically extended over $O(N)$ elements, and the inverse participation ratio (IPR) is typically of order $O(1/N)$. In the large $N$ limit the IPR can thus written as IPR $\sim i(\lambda)/N$, where the coefficient $i(\lambda)$ determines to what extent is the typical eigenvector of eigenvalues equal to $\lambda$ extended. It is indeed well known that on the edges of the spectrum a certain degree of localization can be present, and this is manifested by the divergence of $i(\lambda)$ at the edges. In turn this implies that the IPR of the lowest eigenstates may have a system size dependence slower than $1/N$ and even tending to a finite value in the large $N$ limit. Given the central physical role of the low-energy excitations, it is very important to understand the localization properties of the lowest eigenstates and whether these properties undergo any relevant change approaching the critical point.

The arguments in Sec. 1.2 suggest the eigenvector components $v_i^\alpha$ would be independent Gaussian variables with a variance given by Eq. (13) if not for the constraint that each element $v_i$ must be orthogonal to $S_i$. This constraint not only reduces the number of degree of freedom from $m$ to $m-1$, but correlates fluctuations, and this has a direct implication on the components fourth moment $\langle |v_i^\alpha|^4 \rangle$. The correct computation of the mean fourth moment

can be done by firstly noticing that in the subspace orthogonal to $\boldsymbol{S}_i$, spanned by the vectors $\boldsymbol{u}_{i,a}$, the $m-1$ components of the eigenvector are indeed independent Gaussian variables, and secondly computing the components of the eigenvector in the canonical basis by performing a projection with respect to a randomly oriented spin $\boldsymbol{S}_i$. The prediction for the bulk IPR is thus

$$\text{IPR} = \frac{3(m^2-1)}{Nm^2(m+2)} \int \frac{P_m(H)dH}{|H+G_0-\lambda-G(\lambda)|^4} . \tag{20}$$

The value of the integral in Eq. (20) approaching the lower band edge can be estimated with considerations similar to the ones used for the spectrum, which are detailed in the Appendix. In the paramagnetic phase ($\epsilon > 0$) the integral is dominated by the singularity in $H = 0$ and we have

$$N\,\text{IPR} = i(\lambda) \propto \epsilon^3 \left(\frac{\lambda}{\epsilon}\right)^{-2(m-1)} . \tag{21}$$

Notice that this behavior of the IPR as a function of $\lambda$ cannot hold till the minimum eigenvalue of the system, which is of the order $\lambda_{min} \sim N^{-1/m}$. In fact, such a behaviour would imply a divergent $\text{IPR}(\lambda_{min}) \propto N^{1-2/m}$, which is obviously impossible, being the IPR upper-bounded by 1 by construction. In fact, at the edge of the spectrum eigenvectors localize [32] on sites having a very small field $H_i$.

Suppose to order the sites according to growing cavity fields ($H_1 < H_2 < \ldots < H_N$) and remind that for large fields the Hessian is almost diagonal and thus its spectrum is made of groups of $m-1$ almost degenerate eigenvalues $\lambda_i^a \approx H_i$, with $1 \leq a \leq m-1$, and the corresponding $m-1$ eigenvectors are all localized on the $i$-th site. For smaller fields, but still in the paramagnetic phase, the bulk eigenvectors becomes extended, but localization still takes place on the edge of the spectrum (i.e. for $\lambda \ll 1$). Localization can be derived simply from Eq. (13). This equation admits solutions with one condensed component $|\boldsymbol{v}_i(\lambda)| = O(1)$ if the eigenvalues $\lambda_i^a$ are such that $|H_i + G_0 - \lambda_i^a - G(\lambda_i^a)| = O(N^{-1/2})$. Since $G_0 - \lambda - G(\lambda) \approx -\frac{\lambda}{\epsilon}$, for small $\lambda$, this implies that the lowest lying states are organized in multiplets of quasi-degenerate eigenvalues $\lambda_i^a$ directly proportional to the lowest fields in the large $N$ limit [3]

$$\lambda_i^a \approx \epsilon H_i, \qquad a = 1, \ldots, m-1. \tag{22}$$

The same conclusion would be reached computing the lower eigenvalues to second order perturbation theory, which is exact to the leading order.

As for the eigenvectors, observing that for $\lambda$ close to $\lambda_i^a$ one has $m\,G_{ii}(\lambda) \approx \sum_{a=1}^{m-1} \frac{|\boldsymbol{v}_i(\lambda_i^a)|^2}{\lambda_i^a - \lambda}$, combining with eq.(11) we get that in the thermodynamic limit the square modulus of the $i$-th components, $|\boldsymbol{v}_i(\lambda_i^a)|^2$ takes a finite value

$$|\boldsymbol{v}_i(\lambda_i^a)|^2 \approx \epsilon, \qquad a = 1, \ldots, m-1, \tag{23}$$

while all the other components vanish for $N$ going to infinity. As observed, this mechanism for single site localization is thus similar to a Bose-Einstein condensation on 'single particle' lowest energy states.

We also notice that since the $N$ cavity fields are independent variables with $P_m(H) \simeq H^{m-1}/Z_m$, the probability density of the smallest one $H_1$ is the Weibull distribution

$$\mathbb{P}[H_1 = N^{-1/m}u] = \frac{u^{m-1}}{Z_m} \exp(-\frac{u^m}{mZ_m}), \tag{24}$$

---

[3]It can be shown that the typical splitting of the multiplets is of order $N^{-1/2}$, much smaller than the typical spacing between adjacent $i$ values $\lambda_{i+1}^a - \lambda_i^b \sim N^{-1/m}$. The same kind of considerations show that hybridization between different levels $i$ does not occur for large $N$.

whose mean is given by

$$\langle H_1 \rangle_W = (m Z_m)^{1/m} \Gamma\left(1 + \frac{1}{m}\right), \tag{25}$$

and in turn provides the mean value of the lowest eigenvalues via Eq. (22)

$$\langle \lambda_1^a \rangle_W = \epsilon \langle H_1 \rangle_W . \tag{26}$$

At criticality ($\epsilon = 0$) the spectrum concentrates much more on the lower band edge and this changes the behavior of $i(\lambda)$ close to the edge as follows:

$$i(\lambda) \propto \begin{cases} \sqrt{|\log \lambda|/\lambda} & m = 3 \\ |\log \lambda| & m = 4 \\ \text{const} & m > 4 \end{cases} . \tag{27}$$

We argue that now the IPR vanishes in the thermodynamic limit even for the lowest eigenvectors. Inserting the value of $\lambda_1 \sim N^{-2/3}$ (we neglected logs in $m = 3$), one gets that $IPR(\lambda_1) = i(\lambda_1)/N \to 0$ for large $N$ (in particular $IPR(\lambda_1) \sim N^{-2/3}$ for $m = 3$). In this case we do not find condensation, all the eigenvector elements are vanishing in the large $N$ limit, even at the lower band edge. The spin glass transition appears in this sense to be a delocalization transition for the low eigenvalue modes. Excitations become more and more collective as the critical point is approached. These phenomena are the closest we can have to a delocalization transition in a mean field model.

## 2 Numerical results

We present simulations of the $O(m)$ spin glass at zero temperature in a field in the high field paramagnetic phase and at the transition point. We concentrate our numerical study on the Heisenberg ($m = 3$) case.

We minimize the Hamiltonian starting from a random initial configuration and using an over-relaxation algorithm, that has been used with success in other disordered models with vector spin variables [21]. The basic step of the algorithm consists in aligning sequentially each spin $\boldsymbol{S}_i$ to a direction which is obtained as the linear combination of the gradient, $\sum_j J_{ij} \boldsymbol{S}_j + \boldsymbol{b}_i$, and the over-relaxed spin, $\boldsymbol{S}_{i,\parallel} - \boldsymbol{S}_{i,\perp}$, where $\boldsymbol{S}_{i,\parallel}$ and $\boldsymbol{S}_{i,\perp}$ are the spin projections respectively parallel and orthogonal to the gradient (the over-relaxed spin corresponds to the most distant direction that preserves the energy). The algorithm depends on a single parameter $\Lambda \geq 0$ that fixes the level of greediness via

$$\boldsymbol{S}_i^{(new)} = \frac{\sum_j J_{ij} \boldsymbol{S}_j + \boldsymbol{b}_i + \Lambda(\boldsymbol{S}_{i,\parallel} - \boldsymbol{S}_{i,\perp})}{|\sum_j J_{ij} \boldsymbol{S}_j + \boldsymbol{b}_i + \Lambda(\boldsymbol{S}_{i,\parallel} - \boldsymbol{S}_{i,\perp})|} . \tag{28}$$

We stop the algorithm when the mean displacement $\delta = \frac{1}{N}\sum_{i=1}^{N}(\boldsymbol{S}_i^{(new)} - \boldsymbol{S}_i) \cdot \boldsymbol{S}_i^{(new)} = 1 - \frac{1}{N}\sum_{i=1}^{N} \boldsymbol{S}_i \cdot \boldsymbol{S}_i^{(new)}$ is smaller than a given threshold $\delta_{\max}$ that in our simulations is set in the range $[10^{-9}, 10^{-7}]$. A strictly positive over-relaxation parameter $\Lambda$ allows a much better exploration of the configuration space with respect to simple gradient descent ($\Lambda = 0$). It allows to avoid some high energy local minima and gives rise to better performances, both in terms of the minimal value of the energy reached and in convergence time.

We run simulations covering a wide range of values of external field width $\Delta = k\Delta_c$, with $k$ going from 1 to roughly 10, and sizes $N = 2^n \times 64$, with $n = 0, \dots, 5$. For each size $N$, we choose the number of samples $N_s$ such that $N N_s \geq 3 \times 10^6$. Only close to the critical

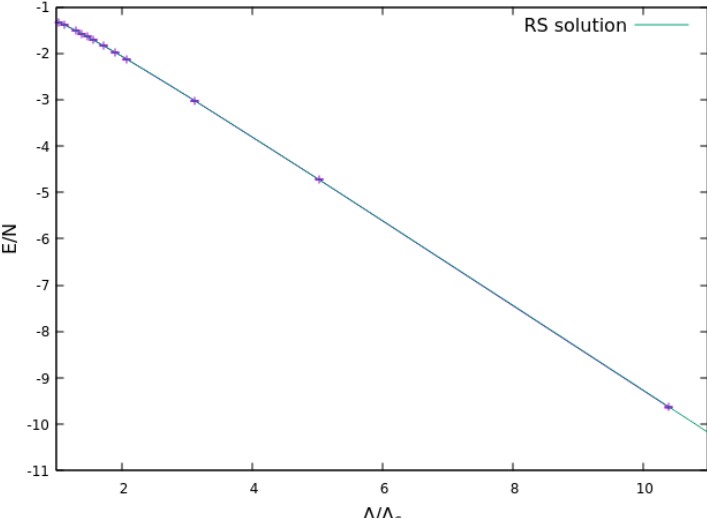

Figure 1: Numerical values of the energy versus $\Delta/\Delta_c$: the continuous line is the theoretical energy density of the model in the paramagnetic phase. The minimization was performed with over-relaxation ($\Lambda \neq 0$) close to the critical point, whereas on the contrary for high values of the field we carried out an ordinary gradient descent minimization.

point we found necessary to set a non-zero $\Lambda$: in particular, we set $\Lambda = 3$ for $k \in [1, 1.5]$, $\Lambda = 2$ for $k \in [1.5, 2]$ and $\Lambda = 0$ for $k > 2$. The convergence time increases when $\Delta \to \Delta_c^+$, signalling that the energy landscape is becoming more complex in this limit. Although our numerical method is heuristic, minimization works very well and we believe that the minima that we reach are very good minima at least in the whole paramagnetic phase including the critical point. Fig. 1 shows a perfect matching between the theoretical ground state energy as a function of $\Delta$ and its value in the simulations.

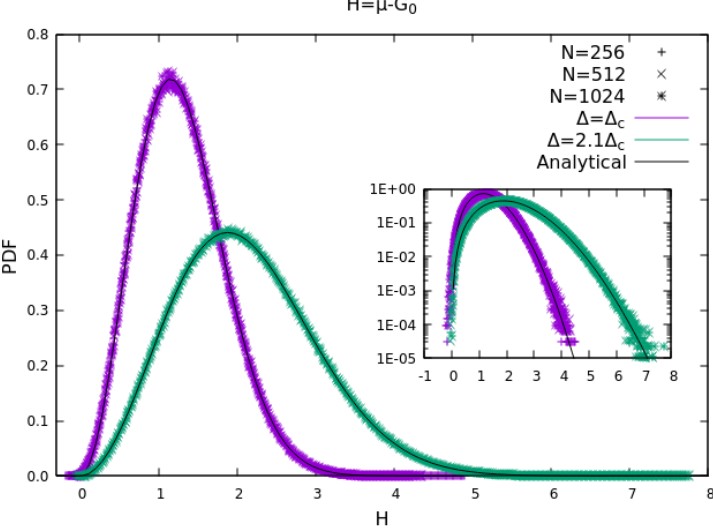

Figure 2: The distributions of the cavity fields for $\Delta = 1.2$ and $\Delta = \Delta_c = 1/\sqrt{3} \simeq 0.577$. The lines report the analytical prediction in the large $N$ limit, Eq. (5). The inset shows the data in a logarithmic scale.

Once reached the energy minimum $S^*$, we compute absolute value of the local fields, $\mu_i = |b_i + \sum_j J_{ij}S_j^*|$, and the cavity fields $H_i = \mu_i - G_0$, where for $G_0$ we use its large $N$ value

$$G_0 = \frac{m-1}{m}\sqrt{\frac{2}{\pi(\Delta^2 + 1/m)}}. \tag{29}$$

In Fig. 2 we show the distribution of cavity fields obtained numerically for two values of the field variance: $\Delta = 1.2$ ($N = 800, 1600$) and $\Delta = \Delta_c$ ($N = 200, 400$). The lines report the analytical result in the large $N$ limit, Eq. (5), and reproduce very well the numerical data.

## 2.1 The Hessian spectrum: bulk and lowest eigenvalues

In Fig. 3 and 4 we present data for the Hessian spectrum $\rho(\lambda)$ obtained with $N = 1024$ and various values of $\Delta$ in the paramagnetic phase and at the critical point. We plot $\rho(\lambda)$ in Fig. 3, with a zoom on the lower edge of the spectrum in the inset. For $\Delta > \Delta_c$, we clearly see the pseudo-gap behavior, $\rho(\lambda) \propto \lambda^2$, which crosses over to the square root behavior at the critical point. To better appreciate the power law behavior for $\lambda \ll 1$, we plot in Fig. 4 the $\rho(\lambda)$ in a double logarithmic scale. The lines are the analytic predictions obtained in the large $N$ limit, that behaves as $\rho(\lambda) \sim \lambda^2$ for $\Delta > \Delta_c$ and as $\rho(\lambda) \sim \sqrt{\lambda}$ at criticality. In the inset we show a scaling plot, that supports the prediction from Eq. (16), $\rho(\lambda) \sim (1-\epsilon)^{3/2}\lambda^2/\epsilon^3$ on the left tail of the spectrum.

In Fig. 5 we single out the spectrum at the critical point plotting it for several values of $N$. The agreement with the theoretical curve is very good. The inset, showing $\rho(\lambda)/\sqrt{\lambda}$, allows to see the formation of the predicted logarithmic singularity for the $m = 3$ case.

We now turn to the statistics of the lowest eigenvalues. We start with a clarification about the notation. In the theory we identify the eigenvalues $\lambda_i^a$ with two indices, because in the localized phase and in the large $N$ limit, the eigenvalues form $N$ multiplets of size $m - 1$ each. However, we prefer to order the eigenvalues measured numerically $\lambda_i$ via a single index

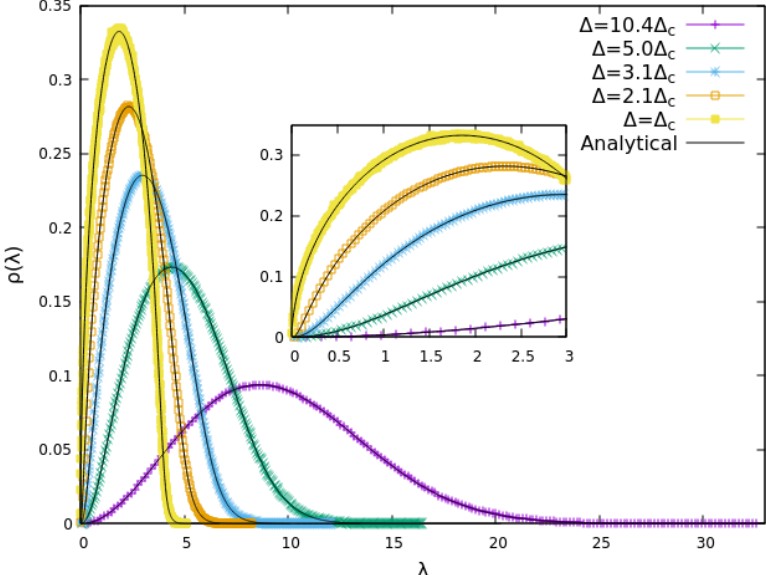

Figure 3: The full spectrum $\rho(\lambda)$ for $N = 1024$ and several $\Delta$ values. The inset is a zoom on the lower edge. The pseudo-gap is clearly visible for large $\Delta$ values. Approaching the critical point the pseudo-gap region shrinks and the curves progressively approach the critical density.

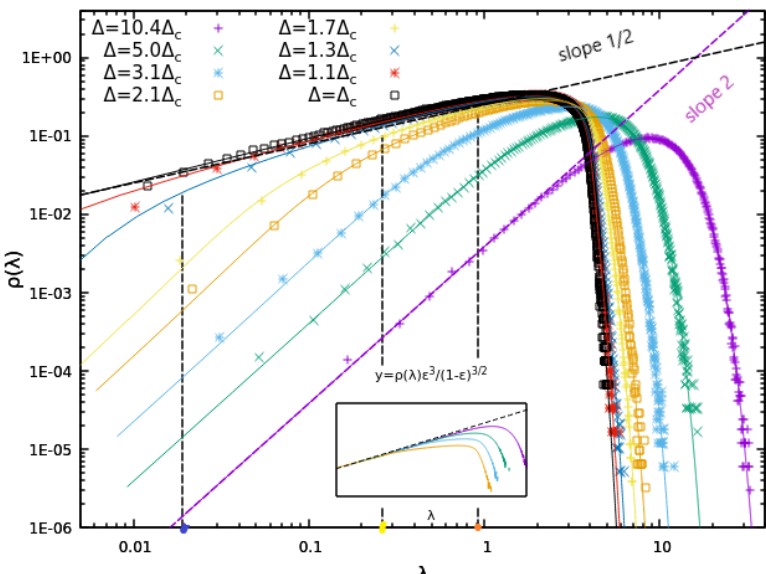

Figure 4: The log-log plot of the Hessian spectrum $\rho(\lambda)$ for $m = 3$ and $N = 1024$ clearly shows the crossover in the behavior at the lower band edge: from $\lambda^2$ at large fields to $\sqrt{\lambda}$ at the critical field. The continuous lines are the analytical spectral densities computed in the large $N$ limit. The dashed vertical lines mark the cross-over values $\lambda^*$ computed in appendix B between the $\lambda^{m-1}$ behavior to the $\sqrt{\lambda}$ one in the curves corresponding to $\Delta/\Delta_c = 1.3, 1.7, 2.1$. The inset is a scaling plot according to Eq. (16).

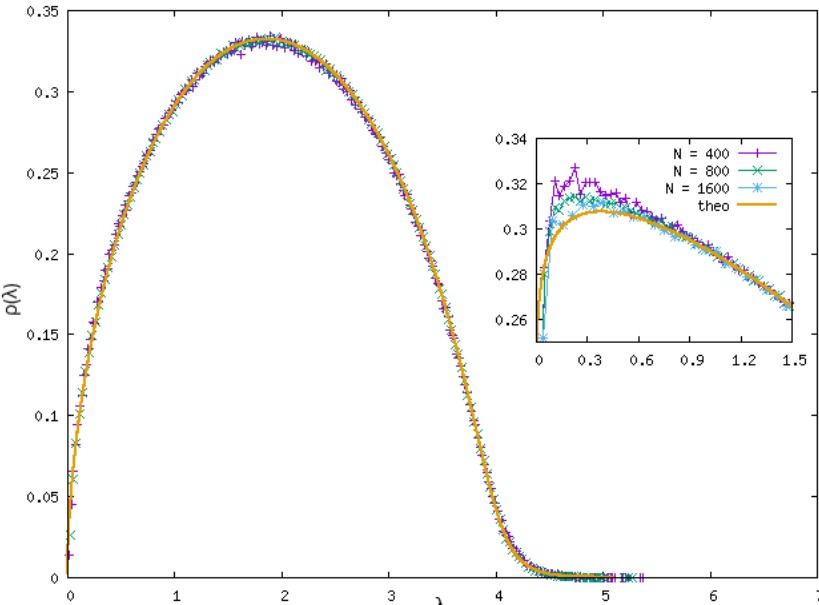

Figure 5: Spectrum of the Hessian at criticality, $\Delta = \Delta_c$, for $m = 3$: data have been measured on roughly a thousand samples of sizes $N = 400, 800, 1600$, while the line is the analytical large $N$ limit. The inset shows $\rho(\lambda)/\sqrt{\lambda}$ to highlight the formation of the logarithmic singularity in the large $N$ limit (orange full curve).

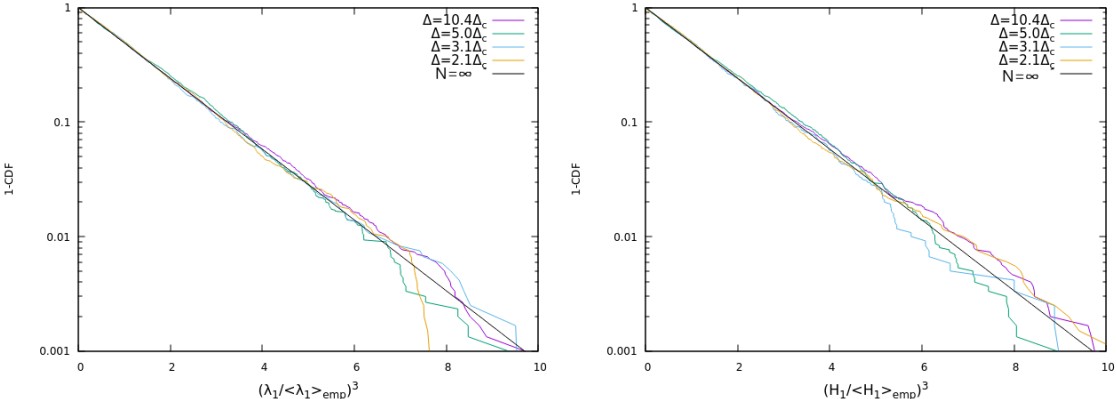

Figure 6: Cumulative distributions for the lowest eigenvalue and lowest cavity field, measured at different $\Delta$ values, follow nicely the theoretical prediction ($N = \infty$).

$1 \le i \le (m-1)N$. Obviously, when eigenstates are very well localized, we will have (e.g. for $m = 3$) that $\lambda_1 = \lambda_1^1$, $\lambda_2 = \lambda_1^2$, $\lambda_3 = \lambda_2^1$, $\lambda_4 = \lambda_2^2$, and so on.

Given that the lowest eigenvalues are expected to scale to zero as $N^{-1/m}$, we study the rescaled lowest eigenvalue $\lambda_1/\langle\lambda_1\rangle_{\rm emp}$ and the rescaled lowest cavity field $H_1/\langle H_1\rangle_{\rm emp}$. The scaling of the empirical averages will be discussed below. According to the theory in the large $N$ limit, see Eqs. (24) and (25), the cumulative distribution of both variables reads

$$\mathbb{P}[\lambda_1^1/\langle\lambda_1^1\rangle_{\rm emp} > x] = \mathbb{P}[H_1/\langle H_1\rangle_{\rm emp} > x] = 1 - \exp\left[-\Gamma\left(1 + \frac{1}{m}\right)x^m\right]. \tag{30}$$

In Fig. 6 we report these cumulative distributions for the largest size $N = 2048$ and several values of $\Delta$. We see an very good agreement with the theory, although we have some noise on the table due to the finite number of samples in our statistics. We will see in Subsection (2.3) that most of the finite size corrections are condensed in small dependence on $N$ of $\langle H_1\rangle_{emp}$ and a much larger one of $\langle\lambda_1^1\rangle_{emp}$. Notice that the cavity fields $H_i$ are independent from site to site; the finite size corrections to the distribution of their minimum could be analyzed e.g. with the RG-like formalism developed in [38].

We check now the relation $\lambda_i^a \approx \epsilon H_i$ for the lowest eigenvalues and cavity fields. We first notice that such a relation is fully compatible with the scaling $\rho(\lambda) = P_m(\lambda/\epsilon)/\epsilon$ we have already checked. As before, we use the rescaled variables $\lambda_1^1/\langle\lambda_1^1\rangle_{\rm emp}$ and $H_1/\langle H_1\rangle_{\rm emp}$ to compare data from different sizes on the same plot. Scattered plot of these two variables for several sizes are shown in Fig. 7 for three values of $\Delta$. It is apparent that for all values of $\Delta$ the two variables are strongly correlated and the correlation improves upon increasing the system size, thus supporting the exact relation

$$\frac{\lambda_1}{\langle\lambda_1\rangle_{\rm emp}} = \frac{H_1}{\langle H_1\rangle_{\rm emp}}, \tag{31}$$

in the large $N$ limit. We notice that the above result, that holds sample by sample, is stronger than the one reported in [32], where the relation is proved only in distribution sense.

The relation $\lambda_1 = \epsilon H_1$ finally follows from the study of the empirical mean values that should satisfy the equality $\langle\lambda_1\rangle_{\rm emp} = \epsilon\langle H_1\rangle_{\rm emp}$. We discuss this relation below when finite size effects are analyzed. Indeed the convergence of the empirical mean $\langle\lambda_1\rangle_{\rm emp}$ to the large $N$ value $\langle\lambda_1\rangle_W$ has finite size corrections that become important approaching criticality (similarly to the broadening of the clouds in the scattered plot in Fig. 7).

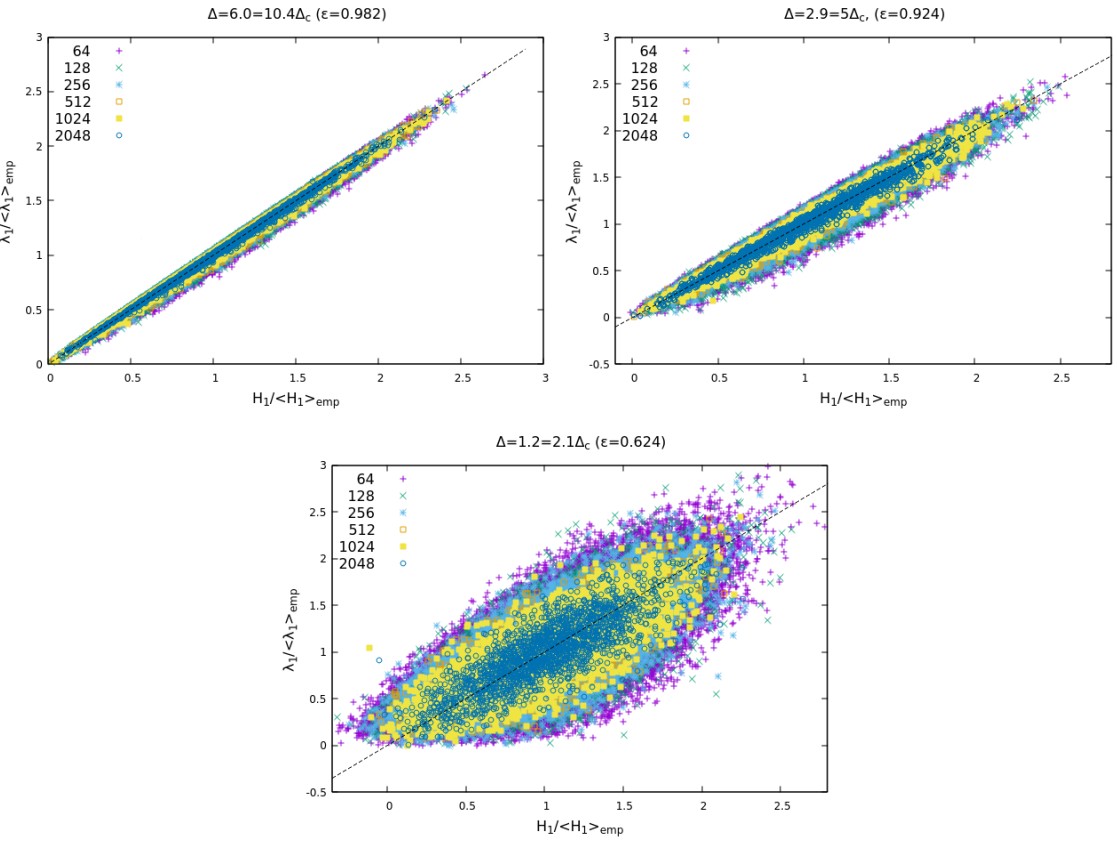

Figure 7: Scatter plots of $\lambda_1^1/\langle\lambda_1^1\rangle_{\text{emp}}$ versus $H_1/\langle H_1\rangle_{\text{emp}}$ for $\Delta = 6.0, 2.9, 1.2$ (from top to bottom) show that these variables become more correlated increasing the system size. Finite size effects (signaled by the width of the clouds) become more evident approaching the critical field $\Delta_c$.

## 2.2 IPR and delocalization transition in the lowest eigenvectors

Similarly to the case of eigenvalues, we observe that the analytical predictions are very well respected for the bulk eigenvector statistics both in the paramagnetic phase and at the critical point. In Fig. 8 we show the sample averages of $i(\lambda) \equiv N$ IPR versus the sample averages of the eigenvalues $\lambda_i$, which show excellent agreement between theory and simulations in the bulk, where the IPR scales as $1/N$.

As already discussed above, the numerical data can not follow the bulk law for $i(\lambda)$ until the lower edge $\lambda = 0$, otherwise the IPR would violate the upper bound IPR $\leq 1$. Indeed, in Fig. 8 (upper panel) we clearly see the deviation from the bulk law for the lowest eigenstates. It remains to understand whether the lowest eigenvectors, i.e. the eigenvectors corresponding to the lowest eigenvalues, are localized and to what extent in the paramagnetic phase, and more importantly what happens approaching the critical point. In Fig. 9 we show the IPR of the smallest eigenvector versus $1/N$. We clearly see that $\text{IPR}_1$ tends to a constant value for $\Delta > \Delta_c$ (although approaching the critical point, the finite size corrections becomes important and the crossover to a constant value will happen for larger sizes).

At criticality, $\Delta = \Delta_c$, a delocalization transition of the lowest eigenvectors takes place and the $\text{IPR}_1$ decays to zero as $N^{-2/3}$ (the exponent is the same one as in a GOE random matrix, given that the spectrum has the same $\sqrt{\lambda}$ singularity).

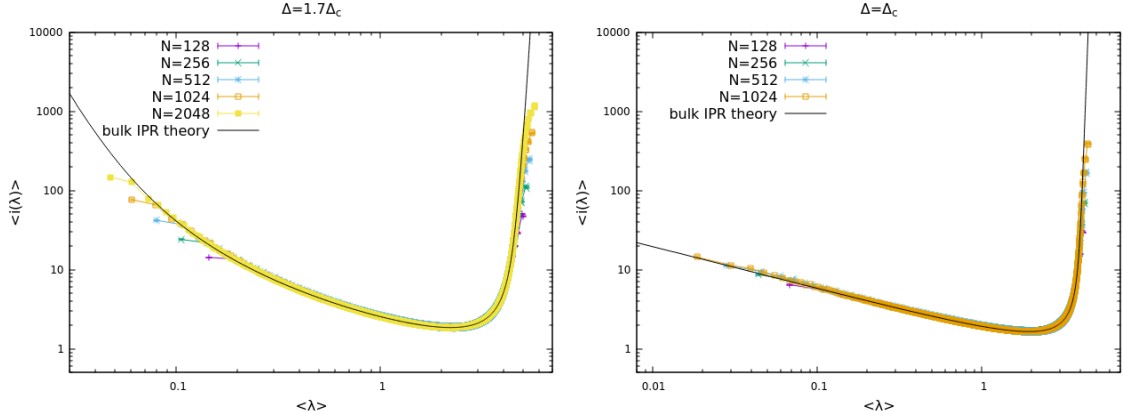

Figure 8: Plot of the sample average of $i(\lambda)$ versus the sample average of $\lambda$, for $\Delta = 1.0 \simeq 1.7\Delta_c$ and $\Delta = \Delta_c$.

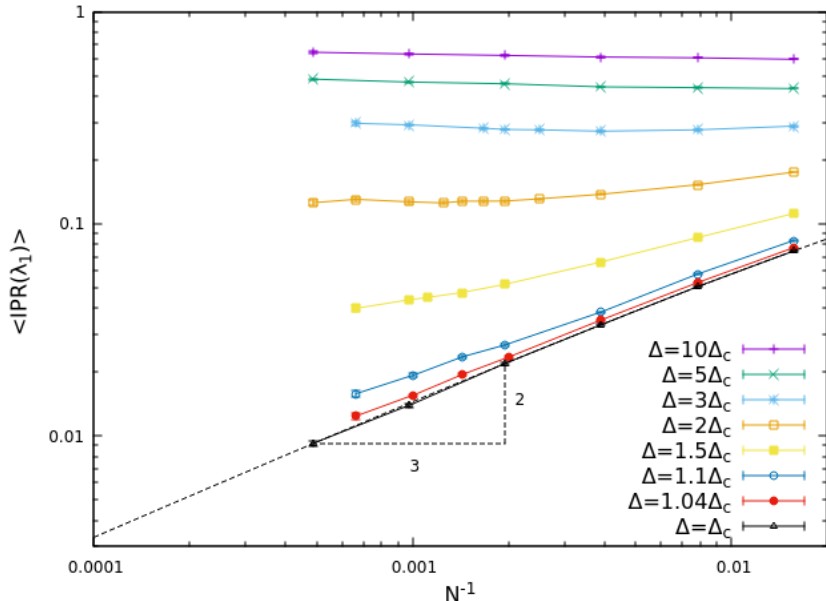

Figure 9: The IPR of the lowest eigenvector versus $N^{-1}$ for several values of $\Delta$. For $\Delta > \Delta_c$ the IPR converges to a finite value, signalling a localization on sites with the smallest external field. At the critical point a delocalization transition takes place, and the IPR decays to zero as $N^{-2/3}$.

## 2.3 Finite size corrections

The analytic theory derived via the cavity method (which actually turns out to be equivalent to the study of a random matrix) is expected to be correct in the whole phase with $\Delta \geq \Delta_c$. Nonetheless we have clearly observed the increase of finite size effects approaching the critical point. This is a standard crossover in critical phenomena, and must be considered with care in order not to make wrong assessments. In this section we analyze numerically the main finite size effects, in order to avoid confusing them with the genuine physical behavior that should eventually dominate in the large $N$ limit.

We start from the study of the empirical averages of the lowest eigenvalues and the lowest cavity field. In Fig. 10 we show the empirical average (over samples) of the lowest cavity

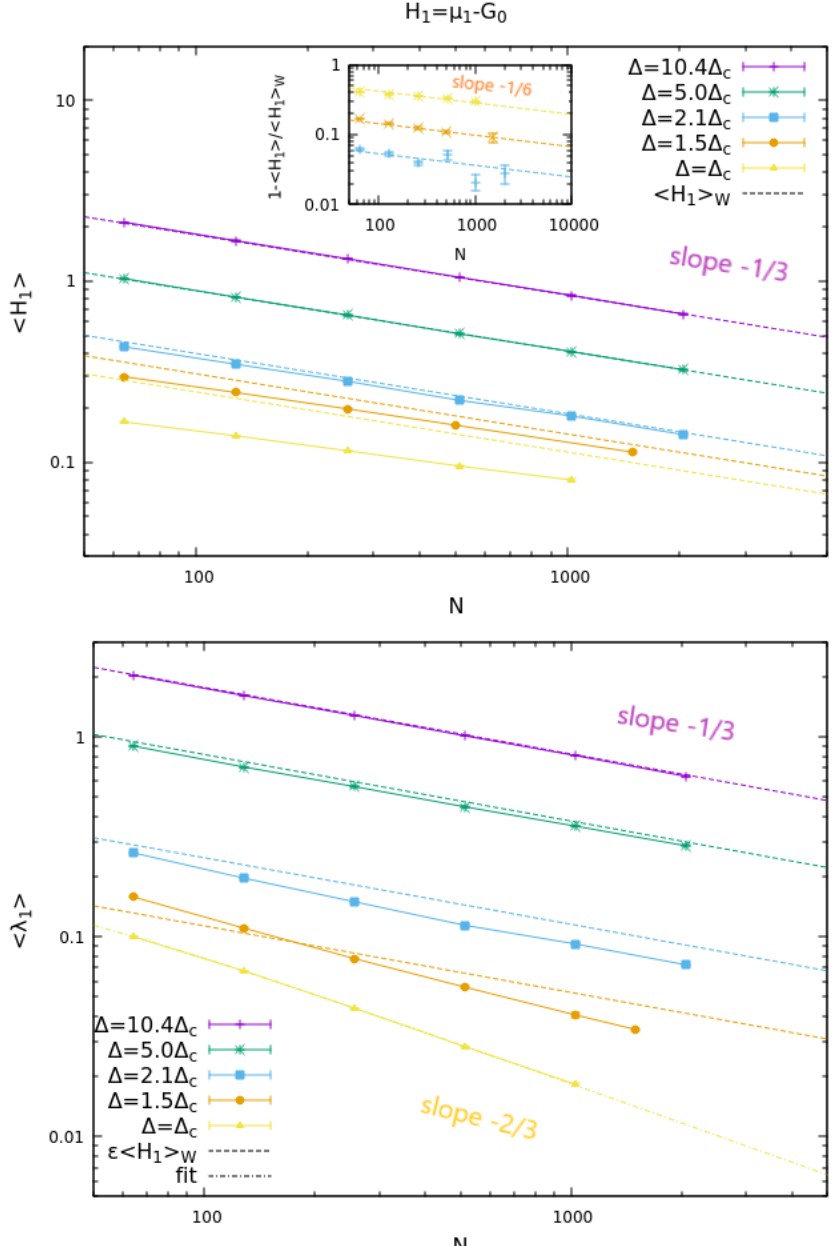

Figure 10: Sample averages of the smallest cavity field $H_1$ (top) and the smallest eigenvalue $\lambda_1$ (bottom) as functions of $N$. For $H_1$ we find a very good agreement with the asymptotic expectation in Eq. (25) for the largest values of $\Delta$, but non-negligible finite size effects close and at the critical point. The inset shows that such finite size corrections goes to as $N^{-1/6}$. For $\lambda_1$, we can appreciate the appearance of a preasymptotic decay as $\Delta \to \Delta_c^+$, that makes very hard to estimate the asymptotic decay close to criticality.

field $H_1$ (top) and the lowest Hessian eigenvalue $\lambda_1$ (bottom). Straight lines are the analytical predictions in the large $N$ limit.

The smallest cavity field shows some deviations from the theory only very close to $\Delta_c$. This deviations are essentially due to fluctuations in the left tail of the distribution of the cavity fields: these fluctuations systematically produce a mean value smaller than the large $N$ limit. Nonetheless the inset shows that the relative difference between empirical average and

theoretical prediction is going to zero in the large $N$ limit.

Finite size corrections in the smallest eigenvalue of the Hessian are more important. In particular, for the values of the field closer to criticality, $\Delta \gtrsim \Delta_c$, we notice a change of slope, due to a crossover from a preasymptotic behavior where $\langle \lambda_1 \rangle \sim N^{-2/3}$ (as if the system where critical) to the asymptotic behavior $\langle \lambda \rangle \sim N^{-1/3}$. This crossover effect is particularly important as it could induce an incorrect estimate of the exponent if the preasymptotic effects are not taken into account.

The origin of this crossover can be well understood looking at the $\rho(\lambda)$ in Fig. 4 and noticing that for $\Delta \gtrsim \Delta_c$ we have (ignoring log factors)

$$\rho(\lambda) \sim \begin{cases} \lambda^2/\epsilon^3 & \text{for } \lambda < \lambda^* \\ \sqrt{\lambda} & \text{for } \lambda > \lambda^* \end{cases}, \tag{32}$$

where $\lambda^* \sim \epsilon^2$ (more details in Appendix C). The asymptotic behavior for the smallest eigenvalue is visible only for sizes such that $\lambda_1 < \lambda^*$, and given that $\lambda_1 \sim \epsilon N^{-1/3}$, we have a crossover size $N^* \sim \epsilon^{-3}$. This is a very strong divergence that actually makes finite size effects dominant in a broad range of fields in the paramagnetic phase.

In Fig. 11 we show the lowest four rescaled eigenvalues $N^{1/3}\lambda_i$ with $i = 1, \ldots, 4$ for $\Delta = 5\Delta_c$ and we make several interesting observations. First of all, these four eigenvalues clearly form two pairs, $(\lambda_1, \lambda_2)$ and $(\lambda_3, \lambda_4)$. The separation between these two pairs is very neat thanks to the large value of the field. Approaching criticality, the separation between pairs becomes less clear. Nonetheless, what it is important is the scaling with $N$ of the separation within a pair and between pairs. These separations are shown in the inset and decay as $\langle \lambda_2 - \lambda_1 \rangle \sim N^{-1/2}$ within a pair and as $\langle \lambda_3 - \lambda_1 \rangle \sim N^{-1/3}$ between pairs. This holds also for lower $\Delta$ values, but one needs to go to larger $N$ values to clearly separates the pairs of eigenvalues.

In the rescaled eigenvalues $N^{1/3}\lambda_i$ the splitting within a pair becomes a finite size correc-

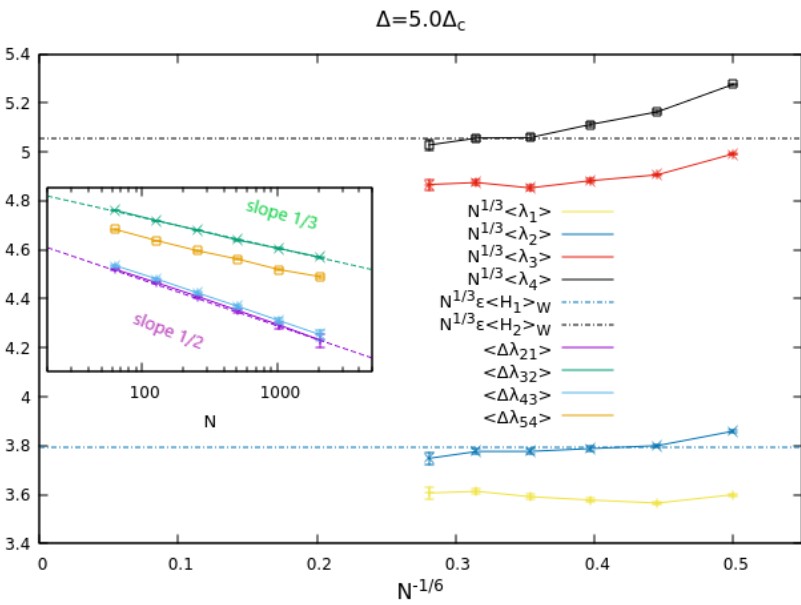

Figure 11: The sample means of the four smallest rescaled eigenvalues $N^{1/3}\lambda_i$ converge to a constant in the large $N$ limit, as they should. The differences within a pair scale as $N^{-1/2}$, while differences between pairs scale as $N^{-1/3}$ (see the inset), so each pair converges to a unique value with $N^{-1/6}$ corrections (hence the horizontal scale).

tion of order $N^{-1/6}$. This is the reason why in the main panel of Fig. 11 we have plotted the rescaled eigenvalues versus $N^{-1/6}$. We notice *en passant* that within each pair of eigenvalues the largest seems to have weakest finite size correction and it is very well compatible with the large $N$ limit (dashed horizontal lines).

Finite size corrections of order $N^{-1/6}$, that is of order $N^{-1/2+1/m}$ for generic $m$ values, seem to be widespread in this kind of models (and also in the related random matrix ensemble). Unfortunately, being the exponent $1/6$ rather small, these corrections persist to very large sizes and may lead to wrong estimation if a theory is missing. We end this section on finite size corrections, by showing how important these corrections may become approaching the critical point.

In Fig. 12 (top) we report the relative difference between the empirical mean and the large $N$ limit, $1-\langle\lambda\rangle_{\text{emp}}/\langle\lambda\rangle_W$, for the lowest eigenvalue and three different values of $\Delta$ as a function of $N^{-1/6}$, which is the expected order of magnitude of the leading corrections. Although the data can be extrapolated to zero for large $N$, thus supporting the validity of the theory in that limit, we also notice that for smaller $\Delta$ values the corrections are severe.

We also notice in the central panel of Fig. 12 that the relation $\lambda_1 = \epsilon H_1$ has much weaker fluctuations with respect to the single averages of $\lambda_1$ and $H_1$.

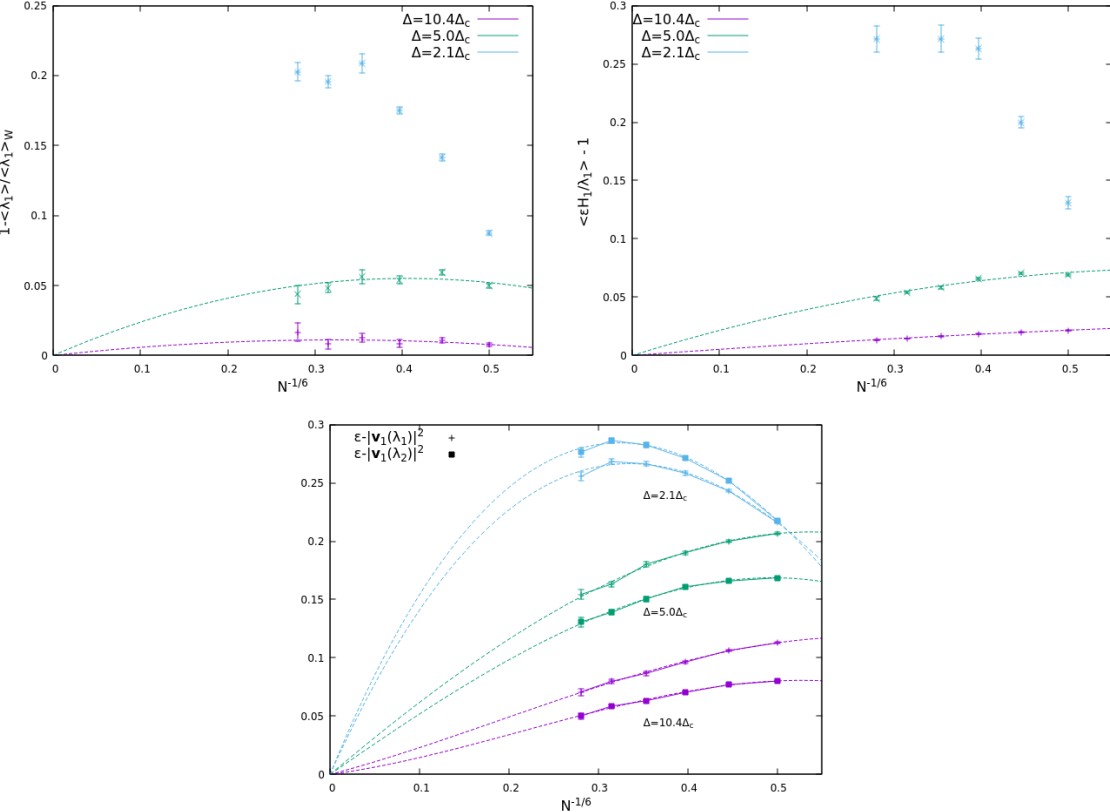

Figure 12: Plots of the relative differences between empirical and asymptotic average quantities: $1 - \langle\lambda\rangle_{\text{emp}}/\langle\lambda\rangle_W$ (above), the relation $\langle\epsilon H_1/\lambda_1\rangle - 1$ (middle) and the largest component of the smallest eigenvectors $\epsilon - \langle|\mathbf{v}_1(\lambda_1)|^2\rangle$ and $\epsilon - \langle|\mathbf{v}_1(\lambda_2)|^2\rangle$ (bottom). We plot these differences as a function of $N^{-1/6}$ which is the expected leading finite size correction. All the data are compatible with a zero value in the large $N$ limit, but finite size effects become very severe as $\Delta$ is lowered, showing a non-monotonic dependence of the curves with respect to the size $N$. Dashed lines are quadratic interpolations to the data.

In the lower panel of the same Fig. 12 we show the difference between the theoretical prediction for the largest component of the lowest two eigenvectors and the actual value measured in systems of finite size. We use the same three values of $\Delta$ as in the other panels. We superimpose a smooth curve showing the data is compatible with a large $N$ limit tending to zero. It is worth noticing that for smaller values of $\Delta$ the extrapolation would be much more difficult giving that finite size corrections are still non-monotonic for the system sizes studied.

## 3   Discussion

In this paper we studied the energy minima of the long-range vector spin-glass with $m$ components in a field. At zero temperature, the model has a paramagnetic phase at high field, $\Delta > \Delta_c$, with a unique isolated minimum and a replica symmetry breaking spin-glass phase with many quasi degenerate minima close to each other at low field. In both phases, the minima are rich of low energy excitations and never show a gap in the Hessian spectrum (at variance with previous claims in the literature).

The Hessian spectrum displays a different abundance of low-energy excitations in the two phases: a pseudo-gap, $\rho(\lambda) \sim \lambda^{m-1}$, in the paramagnetic phase and a square root singularity, $\rho(\lambda) \sim \sqrt{\lambda}$, at the critical point (and probably in the spin-glass phase as well). The square root behavior in $\rho(\lambda)$ is present also for $\Delta \gtrsim \Delta_c$ for not too small eigenvalues, $\lambda > \lambda^*$, and this in turn produces visible finite size effects, and a crossover size $N^*$ diverging at the critical point.

Smallest Hessian eigenvalues are directly connected to lowest cavity fields, which in turn are a consequence of fluctuations in the external random field. Essentially the sites with lowest fields are those where fluctuations can arise more easily and thus produce low-energy excitations. These low-energy modes are quasi-localized in the whole paramagnetic phase and becomes extended only approaching the critical field $\Delta_c$.

The Hessian bulk eigenvectors are random variables, whose components follow Gaussian statistics. As it should, their variance is proportional to $1/N$, but the prefactor depends both on the eigenvalue and on the cavity field, and it is divergent at the lower band edge (i.e. for small $\lambda$) on sites with small field. The lowest eigenvector turn out to be quasi-localized, with its component on a single site remaining finite in the thermodynamic limit. This causes the IPR also to remain finite for any $\Delta > \Delta_c$. At the spin glass transition, low lying excitations become more abundant and minima flatter; as a consequence the lowest eigenvectors are much less localized and the IPR($\lambda_{min}$) vanishes for large $N$.

We would like now to make a general comment about the generality of our results for isolated minima. It is well known that passing from pairwise to $p$-body interaction the phenomenology of spin-glass models becomes the one of Random First Order transition (RFOT) (see e.g. [39]), in particular, one sees the appearance of plethora — actually exponentially many — of isolated glassy energy local minima at different values of the energy. Correspondingly, one sees a stable glass phase appearing in the zero temperature phase diagram. The actual features of the stable glassy minima are very similar to the one of the paramagnetic minima. The vector spin-glass has been indeed generalized to multi-spin interactions in [40, 41] (see also [42]) and display a typical RFOT phase diagram. The computation of the Hessian, its spectrum and the IPR in these minima, follows directly from the present computation for pairwise interactions [43]. Stable glassy minima present features similar to the paramagnetic minima that we have analysed here. In particular one finds a spectral pseudo-gap and quasi-localized excitations.

The absence of spectral gap is generic: it depends on the presence of sites with small cavity field. These, put aside the remarkable exception of spherical models, are deemed to exist even

in random stable phases do to purely entropic reasons in systems with continuous variables (they remind somehow the 'soft spots' in disordered packing [44]). Differently from the common belief, the generic situation of mean-field models is that glassy minima — even the most stable ones — have a spectrum of excitation that extends to zero frequency. Approaching lower and lower frequencies these excitations tend to concentrate on smaller and smaller fractions of the system sites, till for the lowest excitation a single site of the system takes a finite weight of the wave function.

Whether the above scenario is totally due to properties of random matrices [45] or requires some more ingredient, will be discussed in a forthcoming paper.

## Acknowledgments

We acknowledge the support of the Simons foundation (grants No. 454941, S. Franz and No. 454949, G. Parisi). SF is a member of the Institut Universitaire de France. We acknowledge important discussions with A. Scardicchio. We thank G. Tsekenis who collaborated with us at an initial stage of this work.

## A   Analytic derivation of the lower band edge spectrum in the paramagnetic phase

In the following primed quantities will indicate real parts and double primed imaginary parts of complex variables. In general, the solutions to Eq. (12) will be complex if $\lambda$ lies in the spectrum of the Hessian. Let us define then $x = G_0 - G(\lambda) - \lambda = x' + ix''$ and $\widetilde{P}(H) = (1 - 1/m)P(H)$. Detailing the real and immaginary part of Eq. (12), we have

$$G' = \int dH\, \tilde{P}(H) \frac{H + x'}{(H + x')^2 + (x'')^2}\,, \tag{33}$$

$$G'' = \int dH\, \tilde{P}(H) \frac{G''}{(H + x')^2 + (x'')^2}\,. \tag{34}$$

These equations can be easily solved numerically if we know the distribution of the cavity field $H$, in particular this is possible in the paramagnetic phase assuming Eq. (5) for $P(H)$. In this appendix we study analytically the spectral edge. We would like first to illustrate a simple mechanism implying the absence of spectral gap, for any choice of the parameters in the model, and then to show that in the whole paramagnetic phase the spectral density presents a pseudo-gap $\rho(\lambda) \sim x''(\lambda) \sim \lambda^{m-1}$ at small $\lambda$.

We can prove that the spectrum is ungapped with the following argument. From the definition of $x$, we have $x = 0$ for $\lambda = 0$, while $A < 1$ in the whole paramagnetic phase. We should then have $x' = G_0 - G'(\lambda) - \lambda \approx -(1 + \chi_{SG})\lambda < 0$ for small but positive $\lambda > 0$. But in that case, admitting that $x'' = 0$, the resulting integral for $G$ in (33) would be divergent. In order to have a convergent result for $x' < 0$ one clearly needs a small imaginary part $x'' \neq 0$. Let us then proceed to estimate the spectrum in the vicinity of 0. To this aim we observe that defining $\epsilon = 1 - A$, Eq. (12), can be rewritten as

$$\lambda + \epsilon x = -\int dH\, \tilde{P}(H) \frac{x^2}{H^2(H + x)} = -\int dH\, \tilde{P}(H) \frac{x^2 H + x|x|^2}{H^2|H + x|^2} = -x^2 J - x|x|^2 I\,, \tag{35}$$

with

$$J = \int dH\, \tilde{P}(H)\frac{1}{H|H+x|^2}\,, \tag{36}$$

$$I = \int dH\, \tilde{P}(H)\frac{1}{H^2|H+x|^2}\,,$$

giving

$$I|x|^2 = -\epsilon - 2x'J\,, \tag{37}$$

$$\lambda = |x|^2 J\,.$$

It is clear that for $\lambda \to 0$, in order to compensate for the $\lambda$-independent term in the first of (37) at small $x'$ and $x''$, the integrals $I$ and $J$ must be dominated by divergent contributions. Using $x''/((H+x')^2 + (x'')^2) \approx \pi\delta(H+x')$, valid for $|x'| \gg x''$ we can estimate the leading behavior of the integrals as:

$$J \approx \pi\frac{\tilde{P}(|x'|)}{|x'||x''|}\,, \tag{38}$$

$$I \approx \pi\frac{\tilde{P}(|x'|)}{|x'|^2|x''|}\,,$$

so that

$$\epsilon = \pi\frac{\tilde{P}(|x'|)}{|x''|}\,, \tag{39}$$

$$J = \epsilon/|x'|\,,$$

$$|x'| = \lambda/\epsilon\,,$$

$$\rho(\lambda) = \frac{m}{m-1}|x''|/\pi = P(\lambda/\epsilon)/\epsilon \sim \lambda^{m-1}/\epsilon^m\,.$$

This analysis is valid as long as $|x'| \ll \epsilon$ and $x'' \ll |x'|$, i.e. $\lambda/\epsilon \ll \lambda^{m-1}/\epsilon^m$ or $\lambda \ll \epsilon^{\frac{m-1}{m-2}}$. As we approach the critical point, when $|x'| \sim \epsilon$ the singular contribution to $J$ would not be divergent any more and the analysis needs to be revised.

## B  Analytic derivation of the lower band edge spectrum at the critical point and in the spin glass phase

At the critical point $\epsilon = 0$, and Eqs. (37) reduce to

$$I|x|^2 = -2x'J\,, \tag{40}$$

$$\lambda = |x|^2 J\,.$$

The contributions that dominates the integrals for $\epsilon > 0$, become small for $\epsilon = 0$, In particular, one has here $|x'|^{m-2} \ll x''$. The behavior depends now on the value of $m$. Let us start at large $m$. If $m > 4$, one can assume, and check at the end that $|x'| \ll x''$. If this is the case $I$ and $J$ remain finite and the leading contribution is just given by

$$J \approx \int dH\, \tilde{P}(H)\frac{1}{H^3}\,, \tag{41}$$

$$I \approx \int dH\, \tilde{P}(H)\frac{1}{H^4}\,, \tag{42}$$

which are convergent respectively for $m > 3$ and $m > 4$. We therefore have $|x'| \sim (x'')^2 \sim \lambda$ and

$$\rho(\lambda) \approx \frac{\sqrt{\lambda}}{\pi\sqrt{J}}. \tag{43}$$

If $3 < m < 4$ the integral $I$ is divergent in zero and for $|x'| \ll x''$ it gets a contribution of order $I \sim (x'')^{m-4}$, giving $|x'| \sim (x'')^{m-2} \ll x''$. Since $J$ remains finite, the spectral density still behaves as $\sqrt{\lambda}$ in the origin. The cases $m$ equal to 4 or 3 where logarithmic divergences are present should be treated separately. Notice that since $\tilde{P}_m(H) = \tilde{Z}_m^{-1} H^{m-1} e^{-\frac{H^2}{2\sigma^2}}$, we have $I_m = c_m J_{m-1}$. Let us estimate $I_3$, $I_4$. It is clear that $I_3$ is still divergent, and $I_3 \approx 1/(\tilde{Z}_3 x'')$. To estimate $I_4$ we can write

$$J_3 = \frac{I_4}{c_4} \approx -x' I_3 + \frac{1}{\tilde{2Z}_3} \int e^{-\frac{H^2}{2\sigma^2}} d\log(|H + x|^2), \tag{44}$$

which is dominated by the logarithmic contribution in zero:

$$I_4 = c_4 J_3 \approx -\frac{1}{\tilde{Z}_3} \log(|x|). \tag{45}$$

Notice that $I_5 = J_4$ remains finite. As a consequence, for $m = 4$, we can assume $x' \ll x''$ and

$$|x'| \propto -(x'')^2 \log x'', \tag{46}$$
$$J_4 (x'')^2 = \lambda. \tag{47}$$

The logarithmic divergence of $I_4$ does not have consequences for the spectrum that is $\rho(\lambda) \approx \frac{1}{\pi}\sqrt{\frac{\lambda}{J_4}}$. For $m = 3$ we can also assume $x' \ll x''$ and we have:

$$x'' = -2|x'|\log(x'') \approx -2|x'|\log(|x'|), \tag{48}$$
$$\lambda \approx -\frac{1}{\tilde{Z}_3}\log(x'')(x'')^2. \tag{49}$$

Giving $x'' = \pi\rho(\lambda) \approx \sqrt{\frac{\tilde{Z}_3 \lambda}{2|\log\lambda|}}$.

The logarithmic correction to edge-behavior of the critical spectrum at $m = 3$ suggests a different critical solution should be valid for $2 < m < 3$, and the square root behavior of spectrum be modified to an $m$ dependent power. In this case in fact, the singular contribution to $J_m$, namely $J_m^{sing} = \frac{1}{\pi\tilde{Z}}\frac{(x')^{m-2}}{x''}$ would still be divergent if $x'' \sim x'$, which would be therefore a consistent solution. If this is the case we have that $(x'')^{m-1} \sim \lambda$ and $\rho(\lambda) \sim \lambda^{\frac{1}{m-1}}$. Notice that the exponent is larger then $1/2$ for $m < 3$ and is equal to $1/2$ for $m = 3$. Non integer $m$ of course does not have a direct interpretation as a vectorial spin-glass. It is not clear to us if such non-square root critical spectra could be found in some physically realizable glass model.

We can now extend this analysis to the whole spin glass phase. We first notice that our estimates at the critical point only depend on the fact that $A = (1 - 1/m)\langle 1/H^2 \rangle = 1$, i.e. $\epsilon = 0$ and the cavity field distribution behaves as $P(H) \sim H^{m-1}$ in the origin. The first property expresses the divergence of the spin glass susceptibility which also holds in the whole spin glass phase, the second, is a consequence of statistical rotation invariance together with the fact that the fields are gaussian in the paramagnetic phase. In the spin glass phase rotation invariance implies that close to the origin $P(h) \leq Ch^{m-1}$ for some constant $C$. In fact, the cavity field distribution is expected to vanish more rapidly than $H^{m-1}$ in the origin. For example, the analysis in Ref. [27] predicts an essential singularity $P(H) \sim \exp(-a/H)$ for RSB metastable states. Under this conditions the analysis above apply and the integrals $I$ and $J$ remain finite at small $\lambda$. We conclude that the spectral density displays a simple square root behavior in the whole spin glass phase for all $m$.

## C  Crossover from quasi-localised modes to extended modes

For small $\epsilon$, the tail behavior $\rho(\lambda) \sim \frac{\lambda^{m-1}}{\epsilon^m}$ crosses-over as $\lambda$ grows but still $\lambda \ll 1$ to a conventional square root behavior $\rho(\lambda) \sim \sqrt{\lambda - \lambda^*}$. In order to see this, we can rewrite Eq (12) as

$$-\lambda - \epsilon x = x^2 \int dH \frac{\tilde{P}(H)}{H^2(H+x)}. \tag{50}$$

If $m > 3$, the integral appearing in the r.h.s. of (50) is convergent for $x \to 0$, in our region of interest we can therefore estimate it simply as $B = (1 - 1/m) \int dH P(H)/H^3$. We obtain

$$\lambda = (1 - A)x + Bx^2, \tag{51}$$

with $B = (1 - 1/m) \int dH P(H)/H^3$. The apparent gap value $\lambda^*$ is the value of $\lambda$ that make null the discriminant of this equation, namely

$$\lambda^* = \frac{\epsilon^2}{4B}. \tag{52}$$

Correspondingly, the spectral density in the vicinity of $\lambda^*$ can be written as as

$$\rho(\lambda) = \frac{\lambda^*}{\pi \epsilon} \sqrt{\frac{\lambda}{\lambda^*} - 1}. \tag{53}$$

If $m = 3$, we should take into account the logarithmic divergence of the integral in (50). Proceeding as in Appendix A, it is easy to see that $\lambda^*$ gets a logarithmic correction and reads

$$\lambda^* = \frac{3 Z_3}{8} \frac{\epsilon^2}{|\log \epsilon|}. \tag{54}$$

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
