# Peer review of "Delocalization transition in low energy excitation modes of vector spin glasses"

_SciPost Physics, doi:SciPost Phys. 12, 016 (2022)_

## Round 1 · Referee Report · Anonymous (Referee 1) · 2021-9-8

Report

This is a nice thorough paper. The authors have found interesting results in an already well-studied topic. What is surprising is that quasi-localized states exist despite the long-range form of the interactions in the Sherrington-Kirkpatrick (SK) model. This is most surprising.

I expect that there will now be follow-up studies to ascertain the extent to which the results in the SK model extend to sparse graphs and finite dimensional lattices.
  • validity: -
  • significance: -
  • originality: -
  • clarity: -
  • formatting: -
  • grammar: -

Author:  Flavio Nicoletti  on 2021-09-08  [id 1745]

(in reply to Report 1 on 2021-09-08)

Dear reporter, we plan to extend our studies of quasi-localization properties to the same kind of model (m-component spins with random quenched disorder interactions and random quenched external field) on sparse graphs in the forthcoming year. Many thanks for your report!

---

## Round 1 · Referee Report · Anonymous (Referee 2) · 2021-10-2

Strengths

1) New exact results in the thermodynamic limit about a well known spin-glass model.

2) Accurate numerical results and insights into finite size corrections.

Report

In this paper the fully-connected $m$-component vector spin glass model in an external random field is considered and its zero-temperature properties are studied for $m \ge 3$. These models have been studied for more than forty years and are known to exhibit a zero temperature phase transition from a paramagnetic phase at high field and a spin glass phase at low field. In this paper the paramagnetic phase and the critical point are examined most closely, the energy minima in these cases being unique and isolated. In particular, they investigated the eigenvalues ​​and eigenvectors of the Hessian in the minima of the Hamilton operator. With the cavity method correct in the thermodynamic Limes, the authors obtained several interesting results. In particular they studied the eigenvalues and the eigenvectors of the Hessian in the minima of the Hamiltonian. Using the cavity method, which is correct in the thermodynamic limit, the authors obtained several interesting results. The spectrum of the Hessian is found gapless, with different type of pseudo-gaps in the paramagnetic phase and in the critical point. The eigenstates close to the edge of the spectrum show quasi-localisation behaviour. They also found the distribution of the cavity fields and the distribution of the smallest one. The analytical results are confronted with the results of numerical calculations and very strong finite-size corrections are found close to the critical point.

The subject of the paper is interesting, the results obtained could give new impulses to further clarify the behavior of this type of model. The paper is generally well written. I recommend publishing this work. Below I list a few points that the authors should consider.

1) In the abstract the behaviour of the pseudo-gap in the spin glass phase is explicitly written. As far as I can check this is just a conjecture (see the Discussion). I suggest to change this sentence.

2) The crossover from quasi-localised modes to extended modes is studied in Appendix B. Can this result be illustrated on the numerical results in Fig. 4?

3) The distribution of the smallest cavity field is found the Weibull distribution in Eq.(24), which is valid for independent and identically distributed random variables (iidrv). Could the authors comment on this finding?

4) A related question: for iidrv the finite-size corrections to the asymptotic results are known (see: Phys. Rev. E. 81, 041135 (2010)) and recently applied for interacting random systems (see: Phys. Rev. Res. 3, 033140 (2021)). Would it be possible to perform a similar analysis with the data in Fig.6 ?

5) A minor point, a few typos should be fixed. (ipershere -> hypershere; form -> from; be discuss -> be discussed) Also the abbreviation 1RSB-RFOT should be clarified.

---

## Round 2 · Author Response

Dear Editor,

Please find below the resubmittal letter of our paper “Delocalization transition in low energy excitation modes of vector spin glasses”

We thank both referees for the appreciation of our work. We have modified the paper according to the suggestions of the second referee:

"1) In the abstract the behaviour of the pseudo-gap in the spin glass phase is explicitly written. As far as I can check this is just a conjecture (see the Discussion). I suggest to change this sentence."

We added details, in the main text and in the appendix, to our arguments to determine the behavior of the pseudo-gap in the spin glass phase. We hope that the referee finds these arguments convincing and allows us to keep the sentence in the introduction.

"2) The crossover from quasi-localised modes to extended modes is studied in Appendix B. Can this result be illustrated on the numerical results in Fig. 4?"

We have marked the computed cross-over point on three of the curves in figure 4 where the cross-over is more visible. We changed the caption to explain that.

"3) The distribution of the smallest cavity field is found the Weibull distribution in Eq.(24), which is valid for independent and identically distributed random variables (iidrv). Could the authors comment on this finding?"

The cavity fields are indeed iidv with algebraic distribution close to zero, this is the origin of the Weibull distribution of their minimum.

"4) A related question: for iidrv the finite-size corrections to the asymptotic results are known (see: Phys. Rev. E. 81, 041135 (2010)) and recently applied for interacting random systems (see: Phys. Rev. Res. 3, 033140 (2021)). Would it be possible to perform a similar analysis with the data in Fig.6?"

We added a sentence to mention that one could analyse the finite size corrections of the distribution of the minimum field along the lines of Phys. Rev. E. 81, 041135 (2010) (the analysis in the case of the min eigenvalue would be more complicated).

"5) A minor point, a few typos should be fixed. (ipershere -> hypershere; form -> from; be discuss -> be discussed) Also the abbreviation 1RSB-RFOT should be clarified."

We corrected the typos, hopefully all of them. We defined the acronymous RFOT → Random First Order transition, added a reference and avoided 1RSB.

In addition we included a comment about the similarity of Eigenvector Localization in our model and the Einstein condensation in the Bose gas.

Sincerely,
Silvio Franz, Flavio Nicoletti, Giorgio Parisi, Federico Ricci-Tersenghi

---

## Round 2 · List of Changes

Abstract:

"1) In the abstract the behaviour of the pseudo-gap in the spin glass phase is explicitly written. As far as I can check this is just a conjecture (see the Discussion). I suggest to change this sentence."

We added details, in the main text and in the appendix, to our arguments to determine the behavior of the pseudo-gap in the spin glass phase. We hope that the referee finds these arguments convincing and allows us to keep the sentence in the introduction.

Page 8, fig. 4:

"2) The crossover from quasi-localised modes to extended modes is studied in Appendix B. Can this result be illustrated on the numerical results in Fig. 4?"

We have marked the computed cross-over point on three of the curves in figure 4 where the cross-over is more visible. We changed the caption to explain that.

Page 8, fig. 6:

"4) A related question: for iidrv the finite-size corrections to the asymptotic results are known (see: Phys. Rev. E. 81, 041135 (2010)) and recently applied for interacting random systems (see: Phys. Rev. Res. 3, 033140 (2021)). Would it be possible to perform a similar analysis with the data in Fig.6?"

We added a sentence to mention that one could analyse the finite size corrections of the distribution of the minimum field along the lines of Phys. Rev. E. 81, 041135 (2010) (the analysis in the case of the min eigenvalue would be more complicated).

Page 4:

In addition we included a comment about the similarity of Eigenvector Localization in our model and the Einstein condensation in the Bose gas.

Page 15:

We included the expression for the crossover at m=3.

---

## Editorial Decision

published